# Pulmonary Embolism Response Teams: Theory, Implementation, and Unanswered Questions

**DOI:** 10.3390/jcm11206129

**Published:** 2022-10-18

**Authors:** Antoine Bejjani, Candrika D. Khairani, Umberto Campia, Gregory Piazza

**Affiliations:** 1Thrombosis Research Group, Division of Cardiovascular Medicine, Department of Medicine, Brigham and Women’s Hospital, Harvard Medical School, Boston, MA 02115, USA; 2Division of Cardiovascular Medicine, Department of Medicine, Brigham and Women’s Hospital, Harvard Medical School, Boston, MA 02115, USA

**Keywords:** catheter-directed therapy, length of stay, pulmonary embolism response team, pulmonary embolism

## Abstract

Pulmonary embolism (PE) continues to represent a significant health care burden and its incidence is steadily increasing worldwide. Constantly evolving therapeutic options and the rarity of randomized controlled trial data to drive clinical guidelines impose challenges on physicians caring for patients with PE. Recently, PE response teams have been developed and recommended to help address these issues by facilitating a consensus among local experts while advocating the management of acute PE according to each individual patient profile. In this review, we focus on the clinical challenges supporting the need for a PE response team, report the current evidence for their implementation, assess their impact on PE management and outcomes, and address unanswered questions and future directions.

## 1. Introduction

Pulmonary embolism (PE) represents a significant health care burden worldwide. Despite significant advances in risk stratification and management options, PE-related morbidity and mortality remain high. Ever-expanding options for catheter-based therapy and a paucity of randomized controlled trial data to inform clinical guidelines impose additional challenges on frontline clinicians caring for patients with PE. For the past decade, PE response teams have been endorsed to help address these issues by facilitating consensus among local experts while also tailoring the management of acute PE to the individual patient. This article aims to describe the clinical challenges supporting the need for PE response teams, highlight the current evidence for their implementation, critically appraise their impact on PE management and outcomes, and finally, address unanswered questions.

## 2. Challenges for Clinicians

### 2.1. Mortality and Morbidity

Pulmonary embolism (PE) continues to pose a major clinical challenge globally, with the latest data showing 37,571 deaths resulting from PE in 2019 in the United States [1,2,3,4,5]. In the United States, PE-related mortality declined between 2000 and 2006. However, since then, PE-related mortality has increased in young and middle-aged adults [5]. The incidence of PE has also steadily increased around the world [6]. During the early phases of the COVID-19 pandemic, the incidence of PE in hospitalized patients increased to 16.5% and to 24.7% in patients admitted to the intensive care unit [1].

### 2.2. Risk Stratification

Risk stratification is essential to determine the optimal therapeutic pathways for acute PE [7]. Early risk stratification, including hemodynamic assessment, right ventricular (RV) imaging, and assessment for the presence of other comorbidities, is critical to identify patients at higher risk for PE-related morbidity and mortality. Reperfusion therapy, mainly via systemic fibrinolysis, is indicated for all hemodynamically unstable patients (high-risk PE), barring major contraindication. Intermediate-risk PE represents a more heterogeneous population divided into two subcategories: intermediate-high and intermediate-low risk. Intermediate-high risk is defined by imaging evidence of RV dysfunction and an elevation in cardiac biomarkers (such as troponins) in otherwise hemodynamically stable patients. Rescue reperfusion can be considered in these patients if hemodynamic instability develops [8]. Intermediate-low risk PE encompasses patients who are hemodynamically stable with evidence of RV dysfunction on imaging, an elevation in cardiac biomarkers, or neither but with an increased pulmonary embolism severity index [6]. Low-risk PE includes patients who are hemodynamically stable, with a PE severity index of I-II, and no signs of RV dysfunction on imaging. Although this risk stratification model is helpful to categorize patients, it groups a widely heterogenous pool of patients with different comorbidities and symptomatology at the expense of a more individualized approach.

### 2.3. Selection of Reperfusion Therapies

Over the past few decades, research into modalities for reperfusion has rapidly accelerated in an effort to balance efficacy and risk of major bleeding. Such modalities include full- and half-dose systemic fibrinolysis, ultrasound-assisted catheter-mediated thrombolysis (USAT), other catheter-directed fibrinolytic techniques, mechanical catheter-directed thrombectomy/embolectomy, and surgical embolectomy. For patients with high-risk PE, treatment with systemic thrombolysis has a class I recommendation. Surgical embolectomy is an alternative option for patients who fail fibrinolysis, those with major contraindications to fibrinolytic therapy, and those with clot-in-transit [6]. In patients with intermediate high-risk PE, systemic fibrinolysis has largely been supplanted by catheter-based therapies, especially USAT [6,9,10,11]. Other catheter-mediated therapies exist in the form of mechanical thrombectomy, like the FlowTriever (Inari Medical, Irvine, CA, USA) and the Indigo Aspiration System (Penumbra, Alameda, CA, USA). These non-fibrinolytic techniques are especially attractive in patients with major contraindications to thrombolysis.

### 2.4. Guidelines Gaps

Optimal management of intermediate-high risk patients with PE has not been well-established yet and uncertainty with regards to the best therapeutic option has led to increased heterogeneity in the approach of such patients. Several ongoing randomized clinical trials aim to address this key gap in the literature: PEITHO-3 (NCT04430569), comparing reduced-dose fibrinolytic therapy with standard of care (low-molecular-weight heparin [LMWH]); HI-PEITHO (NCT04790370), comparing USAT to parenteral anticoagulation [12]; and PEERLESS (NCT05111613), comparing the FlowTriever system to any number of catheter-directed thrombolytic techniques. While the results of these trials may provide guidance in the future, currently there is an urgent need for a more organized approach to the care of patients with PE. For this purpose, the creation and implementation of PE response teams have emerged in hospitals across the world to standardize PE management within individual medical centers and across health care systems [6,13]. Most notably, the 2019 European Society of Cardiology guidelines for diagnosing and managing acute pulmonary embolism endorsed the setup of institutional PE response team as a class IIa recommendation [6].

## 3. The Theory behind PE Response Teams

The concept of a PE response team stems from the implementation of the “heart team” approach to managing patients with urgent cardiovascular disorders such as myocardial infarction, stroke, and acute aortic syndromes [14,15]. The heart team approach plays a key role in optimizing patient selection for different treatment options, procedural performance, and follow-up care, while promoting shared decision-making [16]. Although the impact of the heart team on PE-related clinical outcomes has not been well-established yet, there are key theoretical advantages, including gathering input from a variety of clinicians, improving timeliness and coordination of care, increasing access to advanced therapies when appropriate, and presenting learning opportunities to trainees on the team [15]. Similar to multidisciplinary stroke teams, heart teams are now endorsed by guidelines that recommend their development across hospitals [14].

In the acute management of PE, PE response teams leverage the richness of the variety of clinicians that participate in PE diagnosis, risk stratification, and treatment (Figure 1). Risk stratification, as discussed above, is a crucial but nuanced step in decision-making regarding reperfusion vs. anticoagulation alone and the type of reperfusion [14]. Navigating an increasing menu of available treatment options, especially catheter-based techniques, requires an understanding of the strengths and weaknesses of each modality, as well as the individual goals of reperfusion for the specific patient, such as improving hemodynamics or gas exchange. For the long-term management of patients with acute PE, implementing these teams can help initiate appropriate follow-up so that post-discharge screening for post-PE syndrome and chronic thromboembolic pulmonary hypertension are not missed. They can also set the basis for determining the duration of anticoagulation as well as for identifying a potential underlying cause of the PE, such as thrombophilia or cancer.

## 4. Implementation

### Who Should Be Part of a PE Response Team?

Various clinicians take a primary role in the care of PE in different hospitals [17]. They act and interact at different stages in the approach to patients with PE: the initial presentation, hospital management, and post-discharge follow-up. The optimal structure of a PE rapid response team remains unknown and may vary widely across institutions [18,19]. PE rapid response teams may include specialists from emergency medicine, critical care medicine, vascular medicine, cardiology, hematology, pulmonary medicine, radiology, interventional radiology, cardiac surgery, clinical pharmacology and pharmacy (Figure 2). Although PE may also be diagnosed in the inpatient setting, the initial steps in acute PE care, which involve diagnosis and risk stratification, are most often carried out in the emergency department setting [17]. After the diagnosis of PE and the assessment of imaging evidence of RV dysfunction by the emergency medicine team and radiology, the focus moves to therapeutic options for PE. Cardiology, vascular medicine, pulmonary medicine, interventional specialists, and cardiac surgery assess whether reperfusion, such as fibrinolytic treatment, catheter-based intervention, or surgery, is indicated. Endovascular interventions, such as mechanical thrombectomy and USAT, are typically provided by interventional radiology, interventional angiology, or interventional cardiology. If surgical embolectomy is warranted, it is performed by cardiac surgery. Clinical pharmacists and clinical pharmacology also play a key role in ensuring appropriate fibrinolytic dosing to optimize their efficacy and safety. Finally, patients diagnosed with PE will require long-term follow-up with either hematology, pulmonology, cardiology, or vascular medicine [20].

The first challenge in establishing a PE response team is the identification of a group of physicians committed to treating PE and comfortable with a shared decision-making model [21]. Because not every hospital has the same breadth of expertise, tailoring the team’s structure based on the institution’s needs and available resources is essential. Large academic hospitals may opt to include physicians-in-training, such as residents or fellows, to serve as first responders. In contrast, smaller hospitals must rely on their attending physicians and advanced practice providers for this responsibility [21]. When it comes to the appropriate team size, the number of individuals included in PE teams varies across institutions [19]. Small groups may have better team efficiency but at the expense of limiting the involvement of other interested stakeholders. On the contrary, a large group may have restricted efficiency but benefit from multidisciplinary breadth [22].

## 5. Structure of PE Response Team

### 5.1. How

The implementation of a PE response team requires commitment of resources and personnel to establish: (1) a tailored response team design; (2) the consultation mechanism; (3) the rollout; (4) a programmatic evaluation process; and (5) mechanisms for quality improvement (Figure 3). First, in the planning and design stage, institutions need to gain support from healthcare stakeholders, identify and assemble team members, and develop protocols and algorithms to help facilitate the rapid response team operation [15]. Second, before implementation, institutions that wish to start such a team should educate all departments associated with PE care on the PE response team process, such as when and how to activate [21]. Third, institutions should iteratively assess their interdisciplinary teams’ impact, which can be conducted by reviewing cases for educational purposes and/or evaluating outcomes for quality improvement [15]. Fourth, since robust evidence assessing the outcome of this approach is lacking, institutions should also use these assessments to contribute to closing the gap in PE care research. One such strategy for improving evidence gaps in the literature is the establishment of registries and clinical trials focused on the impact of response teams [15,21]. Participation in organizations such as the Pulmonary Embolism Response Team Consortium can assist hospitals who are trying to implement such teams (https://pertconsortium.org/ (accessed on 13 August 2022)). Standardized approaches or algorithms can be integrated into a PE response team to reduce some of the heterogeneity between teams while still allowing for individualized care. For example, PE response teams could integrate the European Society of Cardiology 2019 guidelines for patient and PE classification without hindering the ability of the PE response team to tailor the management as it sees fit.

### 5.2. When and What

Operationalizing a PE response team should span the whole continuum of care, starting from patient identification, team activation, triage, early management, and outpatient follow-up. First, to minimize the number of unnecessary activations, institutions should establish a guideline for what kind of presentation should trigger the activation of the PE response team [23]. When a patient requiring consultation is identified, the PE response team is activated in real-time via a paging system [24]. Then, the patient will be rapidly evaluated, triaged, and risk stratified by a PE response team member. Subsequently, the team holds a teleconference or video conference to decide the optimal therapeutic approach according to guidelines and expert consensus. PE response teams can also assist in in-hospital follow-up; some patients may not improve or worsen initially and may require reassessment for potential advanced therapy later in the course of their admission. Lastly, patients are arranged to have a follow-up in an outpatient clinic post-discharge to ensure optimal management across the continuum of care. Patients who develop persistent PE dyspnea or functional limitation after acute PE can be more easily identified with such close follow-up. Often, the same specialists and experts would be consulted when there are concerns for post-PE syndrome or chronic thromboembolic pulmonary hypertension [25].

## 6. Impact to Date

### 6.1. Benefits

A limited number of studies have assessed the benefits of PE response team implementation. With regards to patient outcomes, the utilization of PE response teams has been associated with lower in-hospital mortality [26], 30-day mortality [26], and 6-month mortality by some studies, but not by others [27,28,29,30,31]. Furthermore, these findings cannot infer any cause-and-effect relationship between use of PE response teams and improved mortality rates given their observational design.

The impact of PE response teams on length of stay (LOS) and cost of care has also been evaluated [31]. Care of PE patients is costly, with an average 3-day hospital LOS cost reported at $8764 [32]. In an observational study at the University of Rochester Medical Center/Strong Memorial Hospital, for instance, after the implementation of a PE response team, PE management in the emergency department was associated with a significantly lower triage to diagnosis time, diagnosis to heparin time, and time from triage to disposition (Table 1) [33]. In another study performed at the National University Hospital of Singapore, the authors reported that the median duration of stay in the intensive care unit decreased from 5 to 2 days after a PE response team was developed [27]. A study conducted at the University of Kentucky observed that the creation of a PE response team was associated with shorter hospital LOS, lower total cost of care for the index hospitalization, and lower 30 days post-discharge readmission rates. It was also associated with lower professional and technical costs [26]. Nonetheless, the inherent limitations of observational studies do not allow for definitive conclusions to be drawn.

As for education, fellows in training can be included on the team to gain further opportunities to increase their expertise in risk stratification and acute management of PE [38]. In a study that evaluated fellows’ perceptions after being involved in a PE response team, fellows had an improved educational experience and particularly increased confidence in identifying and managing sub-massive/massive PE and appropriately treating PE with systemic thrombolysis and surgical embolectomy [37].

Another possible advantage with PE response teams, pertaining to the realm of research, is to study aspects of PE care not suitable for randomized controlled trials, such as the use of extracorporeal membrane oxygenation (ECMO). This modality can be helpful in patients with high-risk PE and circulatory collapse, especially in combination with surgical embolectomy [6,39]. The implementation of PE response teams could offer the chance to develop a prospective multicenter, registry of patients requiring ECMO, and different reperfusion therapies to assess their outcomes [39]. This can help clinicians better understand how and when to use mechanical circulatory support.

### 6.2. Potential Unintended Consequences

Implementing a PE response team requires substantial human resources and equipment investment to ensure that all team members can work seamlessly and effectively to provide care to patients with PE. Institutions with infrastructure already in place may have an advantage in initiating the development of such a team. However, for institutions with limited resources, the initial rollout of PE response teams, such as advertisement and expansion to other divisions and specialties, may carry an additional financial burden [22]. On another note, it is also challenging to receive buy-in from physicians to participate in PE response teams, as there is currently no available method to reimburse participation in interdisciplinary teams and participation is still based on volunteerism [17]. There is also a concern that the development of such teams may lead to physician “deskilling” in the management of acute PE as the clinicians become too dependent on the expertise of PE response teams and fall out of touch with more recent trends in clinical practice and the medical literature [14]. On top of that, with many different experts involved in the group, it is still unclear who will be accountable for any medicolegal responsibility [40], and multiple physicians can be subject to litigation. As for advanced therapies utilization, the results are consistent across studies. In studies performed at Massachusetts General Hospital, the University of Virginia Medical Center, and National University Hospital of Singapore, the use of advanced therapies, particularly catheter-directed therapies, increased, which may lead to an increased overall cost [27,28,31,41].

## 7. Unanswered Questions

Most studies on PE response teams have limitations in that they are either descriptive, nonrandomized, or represent a single institution experience [42,43,44,45]. Nearly all of them were retrospective rather than prospective; thus, their direct impact patient outcomes are unclear [42,43,44,45,46]. Furthermore, although most of the literature shows that the PE response teams were associated with a shorter LOS and a decreased cost of hospitalizations, no difference in hospital LOS and cost was seen at the University of Virginia Medical Center [41]. Finally, although the current guidelines recommend the implementation of a PE response team to facilitate the decision-making of acute PE, the recommendation is based on a low level of evidence [6], emphasizing the need for rigorous studies evaluating their effectiveness.

The wide variety of available treatment options makes personalized therapy selection for patients with acute intermediate and high-risk PE possible, but it needs to be tailored by a team of experts who know its indications and limitations. PE response teams have been proposed to help standardize the care management of these patients. However, despite being intended to typically help standardize care management of patients with intermediate and high-risk PE, not all activations of PE response teams were for such patients. For instance, one study of response team utilization showed that a small number of activations were for patients with low-risk PE with additional comorbidities or clot burden [23]. More research is required to assess which types of patients with PE would benefit most from the intervention of PE rapid response teams.

With the increasing recognition of the burden of PE and the ever-increasing therapeutic options, PE response team is becoming broadly incorporated into PE management worldwide. As more institutions integrate these teams, more data will be collected to help inform treatment decisions and guidelines on best practices. However, there seems to be substantial variability in PE management strategies across institutions implementing this concept. For instance, from data of the Pulmonary Embolism Response Team Consortium, the use of catheter-directed and advanced treatments between institutions varies considerably [17,18,19]. In addition, consistent and robust evidence is not available to demonstrate that PE response teams are associated with improved survival. Further studies looking at larger numbers of institutions and involving prospective studies are needed to confirm these findings.

## Figures and Tables

**Figure 1 jcm-11-06129-f001:**
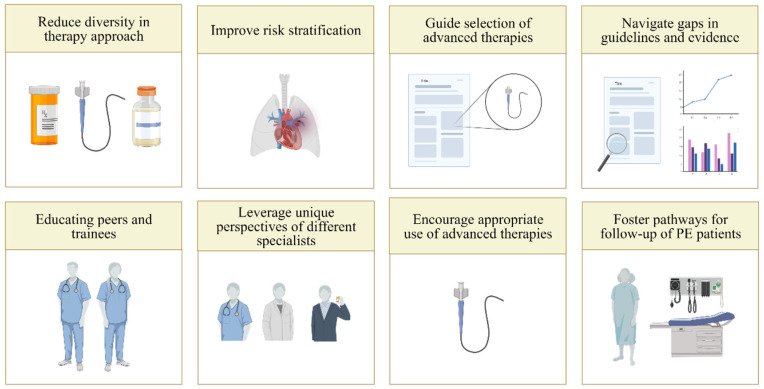
The rationale for pulmonary embolism response teams.

**Figure 2 jcm-11-06129-f002:**
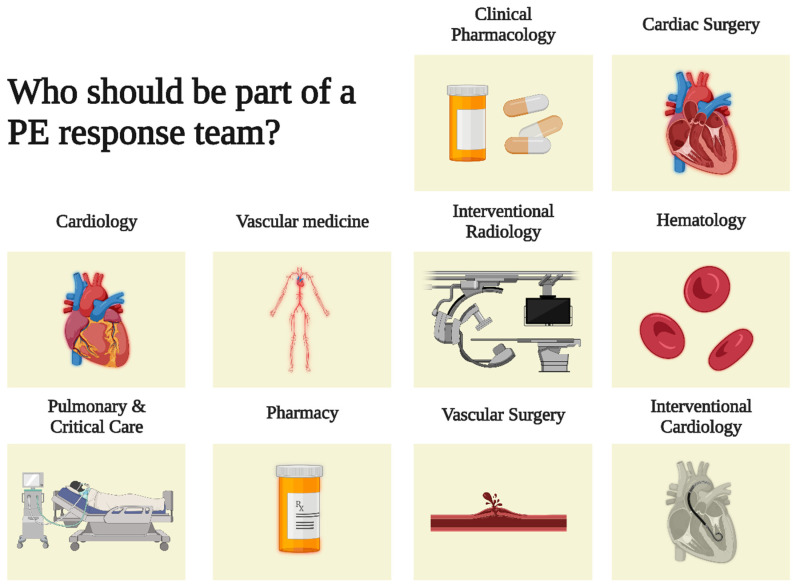
The rationale for a PE response team. PE = pulmonary embolism.

**Figure 3 jcm-11-06129-f003:**
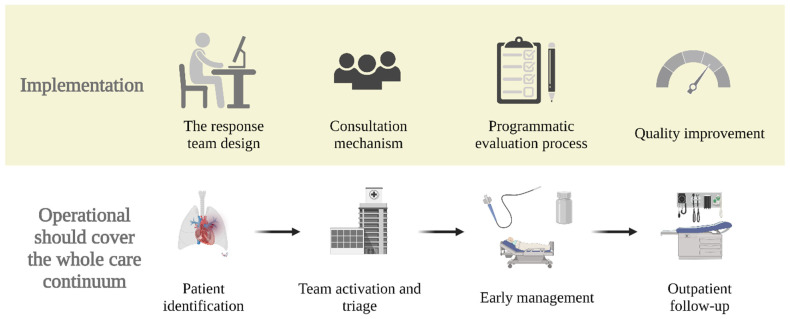
Structure and development of a PE response team. PE = pulmonary embolism.

**Table 1 jcm-11-06129-t001:** Potential impacts and evidence suggesting the role of PE response teams to date [34]. ICU = intensive care unit; LOS = length of stay; PE = pulmonary embolism; RCT = randomized controlled trial.

Impact	Evidence Suggestion Impact	Limitations of the Data
Emergency department efficiency	Associated with lower triage to diagnosis time, diagnosis to heparin time, and time from triage to disposition [33].	Only observational; No RCTs.
Mortality	Associated with a lower 30-day/inpatient mortality, especially in high-risk patients [35].	Observational studies and thus cannot assess causation; Other studies report no change in patient outcomes.
Utilization of resources	Increased use of catheter-directed therapies with no evident increase in bleeding [28,36].	No RCTs.
Length of stay (LOS) and cost of care	One study found no difference in LOS and cost [27]; Other studies reported an association with a shorter length of ICU stay, lengths of hospitalization stay, and cost of care of index hospitalization [26].	No RCTs. Conflicting evidence.
Learning opportunity for physicians in training	Fellows in training may gain more opportunities to increase expertise in risk stratification and acute management of PE [37]; Using prospective institutional registries, fellows in training may use it to conduct their research [38].	No RCTs; Limited data.
Unintended consequences		
Financial cost	Initiation and rollout of PE response teams are costly [22].	
Deskilling of physicians	Referring clinicians may become too dependent on the expertise of PE response teams and deskill them in PE management.	
Medicolegal consequences	Multiple physicians may be listed in a lawsuit because of unclarity of who is accountable for medicolegal responsibility.	

## Data Availability

Not applicable.

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
