# Peer review of "Pulmonary Embolism Response Teams: Theory, Implementation, and Unanswered Questions"

_jcm, 2022, doi:10.3390/jcm11206129_

Round 1

Reviewer 1 Report

Thank you for the opportunity to review this manuscript reviewing the current status and contemporary trends on the management of pulmonary embolism addressing specifically the topic of multidisciplinary pulmonary embolism rapid response teams.

The manuscript is well written and the following are just a few proposals for minor adaptations:

Abstract: Please remove the duplicity of ''worldwide'' in the first sentence.

Risk stratification: Suggest to add definition of low-risk PE for the sake of completeness.

Figure 1: Please correct the typo in ''advanced'' therapies.

Who should be part of a PE response team?: Suggest to also list clinical pharmacology as many hospitals outside the US do not have clinical pharmacy. And, the same for the following sentence on their key role.

Endovascular interventions, such as mechanical thrombectomy and USAT, are typically provided by interventional radiology or interventional cardiology.: Suggest to add interventional angiology or interventional vascular medicine center.

Figure 3: Please correct the typo in ''management''.

When and what: Once MPERRT is set-up, could it also play a role in the management of chronic PE? Often, the same specialists and experts would be consulted in case of PTS or CTEPH. Could authors further elaborate on this point?

Line 277: Please replace ''has become'' by ''is becoming''.

Line 280: suggest to replace ''therapy'' by ''management strategies''.

Reviewer 2 Report

This is a good written review with nice presentation of the concepts surrounding PERTs and nice figures.

The abbreviation MPERRT is not standard. Please use Pulmonary Embolism Response Team (PERT) across the whole manuscript.

Reviewer 3 Report

Pulmonary embolism is a disease with a significant burden of medical care worldwide and of high incidence, which was constantly increasing, especially regained relevance with COVID-19.

From the perspective in which this work is carried out, it makes it interesting, beyond the fact that various revisions of PE have been written, especially with the pandemic.  The importance of multidisciplinary work is a requirement for this type of pathology, as well as given the urgency with which it occurs; being able to avoid them would be the key.

The proposal is interesting, without a doubt a work team is key.  My question would be whether they could perform an algorithm that represents how they would be handled to perform a triage of these patients.

What would a possible organization of this multidisciplinary team look like?
